

# Sedimentary ancient DNA insights into foraminiferal diversity near the grounding line in the western Ross Sea, Antarctica

Ewa Demianiuk[1], Mateusz Baca[2], Danijela Popović[2], Inès Barrenechea Angeles[3], Ngoc-Loi Nguyen[4], Jan Pawlowski[4], John B. Anderson[5], Wojciech Majewski[1]

[1] Institute of Paleobiology, Polish Academy of Sciences, Twarda 51/55, 00-818 Warszawa, Poland
[2] Centre of New Technologies, University of Warsaw, S. Banacha 2c, 02-097 Warszawa, Poland
[3] Department of Geosciences, The Arctic University of Tromsø, 9010 Tromsø, Norway
[4] Department of Paleoceanography, Institute of Oceanology, Polish Academy of Sciences, ul. Powstańców Warszawy 55, 81-712 Sopot, Poland
[5] Department of Earth, Environmental and Planetary Sciences, Rice University, Houston, Texas 77005, USA

*Correspondence to*: Wojciech Majewski (wmaj@twarda.pan.pl)

**Abstract.** Foraminifera are important marine environmental indicators widely used in paleoceanography and paleoclimate studies. They are a dominant component of meiobenthic communities around the Antarctic continental shelf, including rarely studied locations below the ice shelves, close to the grounding line. In this study, we use high-throughput sequencing of sedimentary ancient DNA (*sed*aDNA) targeting foraminifera with two molecular markers, including the ultra-short one newly designed for this study, in five cores from the western Ross Sea, containing sediments up to thirty thousand years old. No foraminiferal DNA is detected in the tills, suggesting a lack of preservation of *sed*aDNA during glacially induced sediment reworking and transport. We reconstruct diverse foraminiferal communities in the open marine settings and significantly less diverse communities in sediments from the slopes of the grounding zone wedges, deposited proximal to the grounding line. Both assemblages are rich in soft-walled monothalamids not preserved in the fossil record and complement the results of earlier micropaleontological studies, allowing a more complete reconstruction of past biodiversity. The newly designed mini-barcode provides higher foraminiferal diversity in surface and subsurface samples than the standard barcode and allows better differentiation between foraminiferal communities in different sediment types. It appears to have great potential for future paleoenvironmental studies, although its taxonomic resolution needs to be evaluated.

## 1 Introduction

The polar regions are the key areas for maintaining the global climate balance (IPCC, 2021) and are major contributors to sea-level rise (DeConto and Pollard, 2016). Although recent climate warming is especially profound in the Arctic, Antarctic ice sheets constitute the largest volume of ice on the planet, totalling almost 60 m of sea level equivalent (IPCC, 2021). Thus, even small changes in the ice sheets volume can significantly contribute to sea level rise. The Ross Sea is a significant drainage outlet for East and West Antarctic ice sheets, which has prompted numerous marine geological investigations, focusing on





behavior of these ice sheets during and following the Last Glacial Maximum (LGM) (e.g., Anderson et al., 2014; Bart et al. 2018; Prothro et al., 2020). These studies have been hampered by the similar composition of sediments deposited in different

glacial and glacimarine settings, widespread re-working of mineral grains, mixing of organic sediment fractions and microfossils, and depleted biotic components in subglacial and near-glacial settings (Domack et al., 1999; Prothro et al., 2018). The use of more advanced analytical methods may help to overcome these problems.

This study focuses on foraminifera, protists that are a major component of meiobenthic assemblages in marine ecosystems. They are abundant, highly diverse, and have short life cycles, making them highly responsive to ecological changes and

therefore particularly valuable for environmental monitoring and paleoenvironmental studies (Jorissen et al., 2009). Due to sensitivity to environmental conditions, i.e., temperature, salinity, pH, redox conditions, or food availability, foraminifera are useful indicators of Antarctic paleoenvironments (Melis and Salvi, 2020; Kilfeather et al., 2011; Majewski et al., 2018, 2020). However, foraminiferal assemblages can be decomposed during early diagenesis, reworking and dissolution due to the strong presence of corrosive bottom waters (Kennett, 1966; Hauck et al., 2012), blurring the full picture of their diversity, and

distorting paleoenvironmental interpretations.

In addition to the relatively robust calcareous and multi-chambered agglutinated foraminifera that are the target of routine micropaleontological studies, there is a diverse group of monothalamid (single-chambered) foraminifera, including allogromiids (Gooday et al., 1996; Majewski et al., 2007). Due to their fragile, mostly organic-walled shells, they are not preserved in fossil and subfossil archives. Their enormous diversity has been revealed by metabarcoding analyses of

sedimentary DNA (Lecroq et al., 2011; Pawlowski et al., 2011; Pawłowska et al., 2014) as well as numerous recent integrative taxonomic studies (e.g., Gooday et al., 2022; Holzmann et al., 2022). Monothalamid foraminifera are particularly well represented in marine restricted environments such as fjords, including environments close to the glacier fronts (Majewski, 2010; Korsun et al., 2023), which was also confirmed by metabarcoding analyses (Nguyen et al., 2023). Only metabarcoding analysis of sedimentary ancient DNA (*sed*aDNA) has the potential to reconstruct their presence in Quaternary deposits.

*Sed*aDNA analysis examines intracellular and extracellular DNA derived from dead cells or shed by living organisms dispersed in the sediment (Pedersen et al., 2015; Torti et al., 2018). Over the last four decades, this method has evolved from analyses of short fragments of mitochondrial or chloroplast DNA (Willerslev et al., 2003; Taberlet et al., 2007), through metabarcoding analyses of amplified DNA fragments using polymerase chain reaction (PCR) or targeted enrichment hybridization (Armbrecht et al., 2021), to broad-based metagenomics analysis covering all groups of living organisms (Slon et al., 2017). The *sed*aDNA

analysis provides new data on past biodiversity or helps to reconstruct paleoenvironmental conditions (Pawłowska et al., 2020a; Armbrecht et al., 2022). Such studies can focus on foraminiferal DNA, improving the reconstruction of their past communities, including rich assemblages of monothalamid foraminifera (Lejzerowicz et al., 2013; Pawłowska et al., 2014, 2020a; Zimmermann et al., 2021; Nguyen et al., 2023).

This study applies the metabarcoding method to analyze foraminiferal communities in sediments deposited in a unique setting

proximal to the paleo-grounding line of the East Antarctic Ice Sheet in the western Ross Sea. Although attempts have been made to assess diversity of Antarctic foraminiferal using metabarcoding (Habura et al., 2004; Pawlowski et al., 2011; Li et al.,





2023), this is the first such analysis completed on subsurface sediments in the Southern Ocean and one of only a few conducted worldwide (Lejzerowicz et al., 2013; Pawłowska et al., 2014, 2016, 2020a, 2020b; Szczuciński et al., 2016). Owing to former studies (Majewski et al., 2020), we combine the micropaleontological data with the newly acquired *sed*aDNA results. The
major goals of this study are (i) to enhance the information on foraminiferal communities in glacier-proximal settings of Antarctic shelf seas, including groups not preserved in the fossil record, i.e., monothalamids and (ii) to assess the possibility of preservation of foraminiferal ancient DNA (aDNA) spread in deposit through glacially-induced remobilization and transport of sediment, and thus the likelihood of carrying on of the aDNA signal from former interglacials to subglacial tills.

## 1.1 Study area

The Ross Sea comprises a vast part of the Antarctic continental shelf. Its drainage area comprises ~25% of both East and West Antarctic Ice Sheets (Rignot et al., 2011), which have been retreating since the LGM (Anderson et al., 2014). The seafloor of the Ross Sea was shaped by glacial fluctuations spanning much of the Cenozoic (Barker et al., 2007). There are sediment-filled troughs, into which ice streams were channelized (Fig. 1). Importantly, these troughs are largely below the lower limit of iceberg scouring, and may therefore provide undisturbed sedimentary archives (Domack et al., 1999). They are separated
by several submarine heights or banks normally shallower than 300 m water depth, bearing signs of intense iceberg scouring (Prothro et al., 2018).

The extent of past ice-sheet grounding lines is marked by grounding zone wedges (GZWs), formed near the grounding-line during periods of relative stability, typically reaching up to 100 m in thickness (Batchelor and Dowdeswell, 2015), and by much less pronounced megascale glacial lineations (MSGL) formed beneath zones of fast-flowing ice (Spagnolo et al., 2014).
These geomorphological features, common in the Ross Sea, are used to reconstruct the grounding line retreat of the Antarctic ice sheets from the outermost continental shelf since the LGM (Halberstadt et al., 2016). In the western Ross Sea, where the East Antarctic Ice Sheet was grounded, subglacial geomorphic features and tills extend to within 30 kilometres of shelf break (Greenwood et al., 2018; Anderson et al., 2014), where embayments in northern parts of JOIDES, Pennell troughs and the Victoria Land Basin (Fig. 1) provided exposure to relatively warm ocean currents.
The Ross Sea continental shelf is the most productive region in the Southern Ocean (Smith, 2022), responsible for > 25% of its total $CO_2$ uptake (Arrigo et al., 2008). The productivity in the Ross Sea is highly seasonal due to variability in solar radiation and sea ice cover and it is only occasionally limited by nutrient depletion (Smith et al., 2014). Diatoms account for about half of this productivity, and are a significant component of Ross Sea surface sediments (Domack et al., 1999).

The Ross Sea is an area of the largest coastal polynya in Antarctica (Park et al., 2018). In such environments, sea ice formation
is accompanied by production of High Salinity Shelf Water (HSSW). This cold, dense, and corrosive to calcium carbonate water dominates the deep basins of the western Ross Sea (Kennett, 1966; Jacobs et al., 1985). It ranges in thickness from 300–400 m to the north to nearly 1000 m in the southern Drygalski and JOIDES troughs. The HSSW is replaced by relatively warm Modified Circumpolar Deep Water, which impinges on to the continental shelf at intermediate depths (Picco et al., 1999) and are capable of winnowing, transporting, and depositing sediment in the inner shelf of the Ross Sea (Prothro et al., 2018). Above





those two water masses, Antarctic surface water is present. Its thickness ranges from just a few tens of meters on the inner
       shelf to ~100 m on the outer shelf (Orsi and Wiederwohl, 2009).

## 1.2 Sedimentological and micropaleontological framework

During the NBP1502A cruise, following a high-resolution multibeam bathymetric survey (Halberstadt et al., 2016; Simkins et
al., 2018), various geomorphological features were cored from different locations with respect to paleo-grounding lines to
better constrain the deglaciation history of the western Ross Sea (Prothro et al., 2020). In this study, we rely on the results of
       this investigation, including the sedimentary facies model, chronological framework (Prothro et al., 2018, 2020), and hard-
       shelled foraminiferal assemblage data (Majewski et al., 2020).

Based on data collected in previous studies (Domack et al., 1999) and more recently, Prothro et al. (2018) used
sedimentological and micropaleontological characteristics to identify sediment facies associated with different environments,
including subglacial, grounding line proximal glacimarine, and open marine. Five cores investigated in this study (Fig. 2)
       recovered sediments deposited in all of these environments, but the entire suite of facies was not sampled for *sed*aDNA. Facies
       1, acquired in the lower parts of cores KC03, KC04, and KC18, which sampled MSGL and GZW topsets, is composed of
       massive diamicton interpreted as till. Facies 2, acquired in cores KC30 and KC49 from the GZW foreset/slope and GZW
       bottomset/toe, is composed of diamicton with variable sorting and interpreted as debris flows initiated from GZW crests close
to the grounding line. Facies 3 is composed of diamicton with abundant granule- to pebble-sized soft sediment clasts. It occurs
       in thin intervals in the middle of cores KC30 and KC49 and is interpreted as the most ice-proximal deposit formed by basal
       melt out of debris-laden ice (Prothro et al., 2018; Simkins et al., 2018). Facies 4, interpreted as meltwater plume deposits,
       occurs in thin intervals in cores KC03 and KC04. Facies 3 and 4 were not sampled for *sed*aDNA. Finally, Facies 5, consisting
       of olive-grey diatomaceous sandy silt, occurs in the uppermost sections of all cores, and records open marine conditions. This
study focused on facies 1, 2 and 5.

Micropaleontological foraminiferal results (Majewski et al., 2020) show that the tills contain predominantly calcareous
foraminifera, including some planktonic forms, at least some of which are clearly reworked. When well preserved and
radiocarbon dated, as in core KC04, foraminifera from tills yielded ages older than the LGM. The open marine sediments are
dominated by agglutinated foraminifera, mainly *Miliammina arenacea* and *Portatrochammina* spp. These are associated with
the presence of HSSW and significant primary production in the sediments. Finally, the glacier-proximal sediments, including



facies formed below ice-shelves, are dominated by calcareous foraminifera, dominated by *Globocassidulina subglobosa* and radiocarbon dated between ca. 11,500 and 23,500 cal yr BP (Majewski et al., 2020).

## 2 Material and methods

### 2.2 Sampling

During the NBP1502A cruise in early 2015, sediment material from five kasten cores was collected. Cores were taken from locations in different morphological features: KC03 - MSGL, KC04 and KC18 - GZW topsets, KC30 - GZW foreset, and KC49 in GZW foreset-toe, penetrating sediments deposited in subglacial, proximal to grounding line, and open marine settings (Table 1 and Fig. 2). The cores were opened immediately after recovery and undisturbed sediment was collected for *sed*aDNA, mostly at a regular depth interval of 40 cm. Two replicates were collected at each sampling depth using disposable laboratory

gloves and sterile spoons to avoid contamination between samples. Sediment samples were frozen at −20°C and transported on dry ice to the Laboratory of Paleogenetics and Conservation Genetics, Center of New Technology University of Warsaw, Poland. The cores were then logged and sampled for grain size, subfossil foraminifera, radiocarbon dating, and water content (Prothro et al., 2018, 2020; Majewski et al., 2020).

### 2.2 Sample preparation and sequencing

Extraction and concentration of total DNA from 34 sediment samples of up to 10 g were conducted with a DNeasy PowerMax Soil Kit (Qiagen), following the producer protocol in a laboratory dedicated to ancient DNA with no prior history of foraminiferal studies. A hypervariable region 37F of SSU rDNA was amplified using two primer pairs specific to foraminifera and amplifying fragments of different length. Firstly, we used forward primer s14F1 (5'-AAG GGC ACC ACA AGA ACG C - 3') (Pawlowski, 2002) paired with reverse primer s15 (5'- CCA CCT ATC ACA YAA TCA TG - 3') (Esling et al., 2015).

To enhance detection of strongly degraded DNA from the reworked material, new forward primer s14F1_SH (5'- GTC CGG ACA CAC TGA GGA TT - 3') was designed and paired with the reverse primer s15, resulting in shorter amplicons, i.e., 19 to 132 base pairs (bp) without primers sequences, with the mean around 64 bp compared to the first pair of primers that amplify fragments of ca. 130 bp (89 to 194 bp). The two fragments are referred in this study as short (SH) and standard (ST), respectively. The new primer was designed with Primer3 (https://pimer3.org) and confirmed in NCBI Primer Blast tool.

The PCR reaction contained 25 µL of AmpliTaq Gold™ 360 Master Mix (Applied Biosystems™), 5 µL of Bovine Serum Albumin, 2 µL of 5 µM primer mix, 15 µL H2O and      3 µL of extracted DNA. After denaturation at 94°C for 5 or 12 min, 60 cycles were applied as follows: 94°C for 20 sec, 52°C for 20 sec, and 72 for 20 sec, with final elongation at 72°C for 2 min. The amplified PCR products were purified on magnetic beads following Agencourt AMPure PCR Purification protocol. Each sample was amplified by both primer pairs in at least five replicates. PCR products were examined by electrophoresis on

agarose gels. Samples with PCR product in four or more replicates were transformed into a double-indexed Illumina





sequencing libraries (Meyer and Kircher, 2010) and sequenced on the MiSeq Illumina platform using MiSeq Reagent Kit v2 2×150 bp.

## 2.3 Bioinformatics

The paired-end raw reads were first quality-checked using the FastQC program (Andrews, 2010). Then, primers and Illumina
tags were removed by Cutadapt (Martin, 2011). Paired-end reads were merged using the "fastq-mergepairs" module and removed putative chimeric sequences using "uchime-denovo" algorithm in VSEARCH v.2.2.2 (Rognes et al., 2016; Edgar et al., 2011), as implemented in the SLIM (Dufresne et al., 2019). Subsequently, the remaining reads were de-replicated, clustered at a 97% similarity threshold into Operational Taxonomic Units (OTUs), and the abundance of OTUs was calculated with "otu-vsearch" module. The non-foraminiferal OTUs (without signature "GACAG"), as well as OTUs with less than 10 reads
in total dataset were removed for further analysis for both ST and SH fragments. To compare the OTU composition between two datasets based on the same region (37F), we first trimmed the ST datasets and identified the shared OTUs using Biopython, and for visualization we prepared Venn diagrams with shared OTUs and reads. OTUs were aligned in BLAST (Altschul et al., 1990) using BLAST best hit search against a curated foraminiferal local database based on minimum similarity (−perc_identity 90%, 7 mismatches or gaps accepted) and minimum coverage (−qcov_hsp 90%) for the taxonomic assignment. The OTUs
below 93% identity were classified at the genus level if possible, or as unassigned foraminifera. For strict taxonomic analysis, to avoid the possible biases, we filtered out the OTUs with less than 10 reads per sample.

## 2.4 Statistics

Before the diversity estimates and statistical analysis, the singleton OTUs (occurring in only one sample) were removed. Statistical analyses were run in R, version 4.1.0 (R Core Team, 2013). All formal hypothesis tests were conducted on the 5%
significance level ($\alpha$ = .05). The OTU tables were rarefied using the lowest read depth corresponding to the sample with the least reads (10 336 for ST and 50 654 for SH). Based on the normalized data, four alpha diversity indexes (i.e., Shannon (H'), Simpson, ACE, and Chao1) were calculated for each sample and compared the distribution of sample diversity across datasets (ST and SH) and environmental settings using the stat_compare_means function of the ggpubr package (Kassambara, 2023). The non-metric multidimensional scaling (nMDS) on the Bray-Curtis similarity coefficient to analyse differences in the beta
diversity of the community composition was calculated with the metaMDS function of the vegan package (Oksanen et al., 2019) with default settings.

## 3 Results

### 3.1 *Sed*aDNA metabarcoding data

Of the 34 samples subjected to *sed*aDNA extraction, PCR products were obtained for 18 samples in at least four replicates
each. These included all surface samples from all cores, representing open marine facies, as well as subsurface samples of




open-marine facies and glacimarine facies proximal to the grounding line for cores KC30 and KC49. No PCR products were observed in till samples (in KC03, KC04, and KC18) (Fig. 2). After quality filtering, merging, and removal of chimeras, non-foraminiferal sequences, and control samples, we obtained 4 253 649 reads, including 1 515 729 reads for the ST and 2 737 920 reads for the SH datasets. As shown in the Venn diagram comparing the same fragments that could be amplified by the SH and ST markers (Fig. S1), 230 OTUs (corresponding to 1 528 719 reads) were shared between the two datasets, with majority, i.e., 55/81% of the OTUs/reads for ST. The SH dataset had 852 unique OTUs representing 79/44% of the OTUs/reads. After further removal of rare OTUs (<10 reads in a single sample, representing <1% of the number of subsampled reads), 1 383 OTUs (397 OTUs of ST and 986 OTUs of SH) representing 4 227 450 (1 511 339 of ST and 2 716 111 of SH; Fig. 5 and Table S1) reads were used for taxonomic analysis. On average, the number of OTUs/reads per sample reached 36/88 902 for 17 ST samples with positive results and 84/135 80 for 20 SH positive samples (Table 2). The ST sample from 200 cm depth in KC49 ST was not included as no OTUs remained after removing OTUs with <10 reads. SH samples in KC30 from 200 and 280 cm were analysed in two replicates.

There are clear differences in the DNA results from different sediment types. No PCR products were obtained from the tills (cores KC03, KC04, and KC18). The number of OTUs strongly graded from the highest in the surface open marine sediments, i.e., 90/209 OTUs for the ST/SH primers on average, through subsurface open-marine (23/77 OTUs) to the lowest 5/19 OTUs in the glacier-proximal sediments below 120 cm in cores KC30 and KC49 (Table 2).

## 3.2 Alpha diversity

Normalized foraminiferal alpha diversity expressed by the ACE and Chao1 indices are both more than twice as high for SH than for ST and show a significant decrease from surface, through subsurface open-marine to glacier-proximal samples (Fig. 3). In contrast, the Simpson index shows roughly similar values for ST and SH for surface and subsurface open-marine samples, at ca. 0.8 for both ST and SH, and significantly lower for glacier-proximal samples, at ca. 0.5 (Fig. 3). The Shannon index is somewhere in between. It is clearly higher for SH than for ST, but not as significantly as in the case of the ACE and Chao1 indices, and shows significantly higher values for the open-marine than for glacier-proximal environments (Wilcoxon tests, **P < 0.01) for both datasets, but the difference between surface and subsurface open marine values is reduced in comparison with the ACE and Chao1 (Fig. 3). For the ST dataset, the Shannon index averages 2.2 for the open marine surface samples and 1.8 for open marine subsurface samples. For the glacier-proximal samples, the Shannon index is significantly lower, reaching only 0.8 on average. For the SH dataset, the Shannon values are significantly higher, averaging 3.2 for open marine surface samples and 2.4 for open marine subsurface samples. Shannon values are again lowest in the glacier-proximal environment, averaging only 0.8.

## 3.3 Beta diversity

The difference in community composition between the two datasets is reflected in the nMDS plots (Fig. 4). In general, SH and ST datasets produced similar patterns but they are more scattered in the SH datasets. For the ST dataset, the foraminiferal



communities in the open-marine samples form clusters slightly different for surface and subsurface samples, while those in the glacier-proximal samples are largely scattered and overlap with some subsurface open marine samples (Fig. 4A). For the

SH dataset, communities in surface open marine, subsurface open marine and glacier proximal sediments form distinct clusters (Fig. 4B). Those from cores KC30 and KC49 cluster separately in the case of the open marine subsurface, actually for both datasets, and in the glacier-proximal samples for the SH dataset only. The surface open-marine samples tend to form tight clusters and overlap between different cores.

### 3.4 Taxonomic composition

The OTUs assigned to the reference sequences represent all major foraminiferal groups (Fig. 5). They are clearly predominant, representing 69.3% and 85.5% of the total ST and SH datasets, respectively. When calculated on average per sample, the assigned OTUs accounted for 78.8% of the ST dataset and as much as 88.3% of the SH dataset (Table S2). The assigned OTUs represent 67 genera and 53 named species (Table S3), representing 33 genera for the ST primers and 59 for SH, and they do not always overlap (Table S3). In addition, OTUs assigned to monothalamids represented 17 different clades (Fig. 7); 12 clades

for the ST primers and 17 for the SH together with 10 other divisions based strictly on environmental sequences called ENFORs groups, nine detected with ST and ten with SH primers, as well as numerous unclassified environmental sequences; see Table S1.

In both datasets, OTUs of Monothalamea dominate overall, reaching an average of 58.5% for ST and 58.4% for SH datasets. The number of monothalamid OTUs is highest in the surface sample of KC49, with up to 92 OTUs in the ST dataset and 278

OTUs in the SH dataset (Figs. 6 and 7). The rotaliids are the second most abundant group in the ST dataset (9.7%). In SH, the most abundant groups after Monothalamea are planktonic foraminifera (12.9%), and Tubothalamea (12.1%). The other taxonomic groups (e.g., Textulariida) are also recorded but in very small numbers (Table S2). Unassigned OTUs average 11.7% in the SH dataset and 21.2% in the ST dataset.

In the ST dataset, the most commonly detected foraminifera, i.e., taxa identified in the largest number of samples, are the

planktonic *Neogloboquadrina pachyderma*, found in 11 out of 17 samples, the monothalamid genera *Micrometula* and *Hippocrepinella* in 9 samples, the rotaliids *Nonionella auris* and *Cibicidoides wuellerstorfi* in 6 samples, and the monothalamid genus *Psammophaga* in 5 samples (Table S1). All of these foraminifera are found in several surface-open marine sites and at least one glacier-proximal site. In the SH dataset, the most common are *N. pachyderma*, found in 18 of 20 samples, the miliolids *Cornuspira antarctica* in 16 samples and *Cornuspiramia* sp. in 11 samples, the monothalamids *Gloiogullmia* sp. and

*Micrometula* sp. in 11 samples, *Hippocrepinella* sp. and *Saccammina* sp. in 7 samples, and the rotaliids *N. auris* and *G. subglobosa*, found in 7 and 6 samples respectively, along with a few clades of monothalamids (Table S1). Of these most commonly identified taxa, only *G. subglobosa* is not found in glacier-proximal samples.





## 3.5 Downcore assemblage variability

The proportions of the major taxonomic groups change with core depth. From the depth of 40 to 120 cm depth, corresponding
to the subsurface open marine system in KC30 and KC49, the number of OTUs is lower than in surface samples, but the
assemblage is relatively diverse, with a strong dominance of monothalamids and unassigned OTUs. From a depth of 160 cm,
corresponding to the glacier-proximal marine samples in those cores, the number of OTUs decreases sharply. For the glacier-
proximal samples, in the ST dataset there is a strong dominance of Monothalamea and an absence of Tubothalamea,
Textulariida and unassigned OTUs (Fig. 6); benthic and planktonic Rotaliida may also be missing in some samples. In the SH
dataset of KC30 samples, Monothalamea is less dominant in glacier-proximal samples below 120 cm than in overlying open
marine samples, and planktonic Rotaliida and Tubothalamea may be more abundant. Textulariida and benthic Rotaliida are
absent below 120 cm. Unassigned OTUs vary from 0 to ca. 30%. In KC49 samples, on the other hand, Monothalamea are
more dominant, especially in the two bottom samples, but Textulariida are consistently absent below 120 cm and benthic
Rotaliida are present only in the bottom sample.
The occurrence of particular OTUs/species is difficult to follow downcore as it is highly irregular. The variability of the overall
assemblage, including the discontinuity below core depths of 120 cm, is shown by different clusters in the nMDS plot for SH
(Fig. 4) and presence/absence in open-marine vs. glacier-proximal facies in Table S3. Except for two monothalamiid species,
*Bathysiphon flexilis* in KC30 at 200 cm depth and *Conqueria laevis* in KC49 at 235 cm depth, both recognized by SH, there
are no assigned foraminiferal taxa specific to glacier-proximal sediments (Table S3). Among Globothalamea and
Tubothalamea, only a few are genetically detected in the glacier-proximal facies (Table S3). With the ST primers it was
possible to amplify *G. subglobosa* and *C. wuellerstorfi* from single samples in KC30 and *N.auris* and *Cornuspiramia* sp.
(Tubothalamea) in KC49. The SH primers revealed the presence of N. auris in the deepest sample of KC49, family of
Trochamminidae (Textulariida) in KC30, and abundant *C. antarctica* (Tubothalamea) and planktonic *N. pachyderma* in cores
KC30 and KC49 (Table S1). Compared to globothalamids and tubothalamids, monothalamids are more abundant in the glacier-
proximal samples. In core KC30, OTUs belonging to the monothalamid clades A, B (*Bowseria* sp.), BM (B. flexilis and
*Micrometula* sp.), C (*Gloiogullmia* sp. and *Hippocrepinella* sp.), D, G, V and the environmental clades ENFOR2 and ENFOR3
are recorded for the ST and SH markers (Tables S1 and S3). In the KC49 samples from the same facies, we observe only three
monothalamid OTUs for the ST marker (*Psammosphaera* sp., *Micrometula* sp. and an OTU belonging to clade G), while for
the SH marker we obtained 43 OTUs. Most of them, 33 OTUs, are from the deepest sample at 235 cm depth, including
numerous OTUs representing the genera *Micrometula*, *Saccamina*, *Gloiogullmia*, and a few OTUs assigned to *C. laevis* and
*Bathyallogromia* sp., as well as clades G, M3 and ENFORs 2, 3 and 4 (Table S1).



## 4 Discussion

### 4.1 Absence of the ancient DNA signal in glacially redeposited sediments

Numerous studies have focused on the marine record of Antarctic deglaciation, particularly in the Ross Sea basin, but the
details of ice sheet behaviour remain uncertain (Anderson et al., 2014; Prothro et al., 2020). This is largely due to widespread reworking of sediments during multiple ice sheet retreats and expansions (Naish et al., 2009) and difficulties in distinguishing between subglacial and glacimarine sediments (Domack et al., 1999; Prothro et al., 2018). The reworking process also affects biogenic carbonates and organic matter (Domack et al., 1999; Prothro et al., 2020), raising the question of whether the DNA signal representing past glacial cycles can be transferred to post-LGM sediments. To determine the potential for DNA
persistence in the harsh and dynamic conditions of the Antarctic shelf, it is necessary to understand the processes that affect DNA in the sediment and its subsequent extraction. DNA preservation is strongly related to environmental conditions such as pH, salinity and temperature, the chemical composition of the sediment, and the biotic activity of living organisms, mainly bacterial communities, which depend on the nature of the nutrient components available in the sediment (Levy-Booth et al., 2007). The factor that significantly increases the possibility of DNA preservation is its ability to bind to the mineral and organic
grains that make up the sediment. This process protects DNA from degradation by microbial activity (Blum et al., 1997; Corinaldesi et al., 2008).

It has been shown that DNA can be bonded in significant amounts on clay minerals due to their large absorption surface (Lorenz and Wackernagel, 1987, 1992; Blum et al., 1997; Levy-Booth et al., 2007; Slon et al., 2017). The binding strength of DNA to clay molecules is significantly higher at pH values > 5 (Levy-Booth et al., 2007). pH values in the Ross Sea water
column average between 7.9 and 8.3 (Rivaro et al., 2014) with similar values in sediments (Li et al., 2019), which together with low temperatures suggests that the general conditions in the sediments studied are favourable for DNA preservation. Calcium carbonate is also known to have a high potential to bind DNA, regardless of pH, but this binding can destroy the structure of DNA molecules (Freeman et al., 2023). The $CaCO_3$ content in Ross Sea sediments is generally low, averaging 2%, and even the highest values are <10% (Hauck et al., 2012), but it may still play some role in DNA binding, especially
locally. Nevertheless, considering the delicate structure of DNA and its rapid hydrolysis (Rawlence et al., 2014), a significant remobilization of its molecules together with strong sediment mixing and re-suspension seems rather unlikely (Willerslev et al., 2004; Armbrecht et al., 2019). However, there is no evidence that sedimentary DNA is affected by less perturbing processes, such as glacier-induced sediment deformation or redeposition of sediment aggregates, that occur near the grounding line (Prothro et al., 2018; Robinson et al., 2021).

In our study, we were unable to amplify foraminifera in subsurface samples from sites located in areas with MSGL (KC03) or GZW topsets (KC04 and KC18) for either the ST or ultra-short SH primer pairs. In addition, spectrometric measurement of DNA content after extraction and concentration failed to detect its presence. Organic carbon and foraminiferal tests from these deposits yielded radiocarbon ages <30 000 years (Fig. 2), so these samples are young enough to allow for the preservation of *sed*aDNA (Lejzerowicz et al., 2013; Pawłowska et al., 2020b; Armbrecht et al., 2022). Consequently, the lack of measurable





DNA could be attributed to sediment mixing and dilution. This is based on the interpretation that MSGL and GZW topsets are products of bed deformation and sediment reworking during long-distance transport beneath the ice sheet (Domack et al., 1999; Spagnolo et al., 2014; Halberstadt et al., 2018; Robinson et al., 2021).

Ross Sea tills are typically over-compacted, stiff diamictons with low pore water content (Domack et al., 1999; Prothro et al., 2018), indicating that these sediments must have been under considerable pressure during ice sheet expansion (Tulaczyk et al., 2001; Robinson et al., 2021). Perhaps this was another factor affecting the preservation of *sed*aDNA. Tills in our samples have
<30% water content, i.e., from 25.4% in KC03, 27.5% in KC04 to 29.3% in KC18 on average (Fig. 2), whereas in overlying open marine sediments water content reaches up to 70% (Prothro et al., 2018). This is a significant difference, but similar values to those recorded in the tills have also been measured in glacier-proximal sediments in the lower parts of KC30 (30.8% on average) and KC49 (28.6%), which revealed the presence of *sed*aDNA, suggesting that compaction was not a critical factor
in preserving DNA.

Overall, it is not clear whether the paucity of DNA in the tills observed in our cores KC03, KC04 and KC18 (Fig. 2) applies to all types of subglacial sediments and all types of DNA material. This needs to be confirmed in further studies of relatively young tills with well-preserved microfossils. The absence of DNA in glacial sediments can be further confirmed, for example, by the target enrichment technique by hybridization method based on a single-strain library, i.e., 30 bp long, which is able to
capture very short *sed*aDNA fragments of a dozen to several tens of bp (Wales et al., 2015), performed not only on foraminifera but also on other marine organisms.

## 4.2 Low foraminiferal diversity in *sed*aDNA samples near the grounding line

Although foraminifera are the key microfossils for reconstructing past paleoenvironments, their distribution in some important Antarctic habitats, especially under ice shelves, is still fragmentary. Recent assemblages have been documented from only two
sites below the Ross Ice Shelf, i.e., the testate forms from about 400 km (Lipps et al., 1979) and the monothalamid foraminifera from 10 km south of the calving front (Pawlowski et al., 2005). Similar studies have been carried out beneath the Amery Ice Shelf in East Antarctica, revealing abundant planktonic foraminifera (Hemer et al., 2007), which, together with other microplankton, increased just prior to the colonization of sub-ice shelf habitats by benthic infauna and filter feeders (Post et al., 2007). These direct observations are rare and cannot provide the baseline data needed for reliable environmental
reconstructions.

An estimate of sub-ice shelf communities can also be based on studies in restricted, low productivity Antarctic conditions, such as Explorers Cove in McMurdo Sound, which is characterized by cold and nutrient-poor waters from beneath the Ross Ice Shelf (Barry and Dayton, 1988) and semi-permanent sea ice (Gooday et al., 1996). The fauna of Explorers Cove is heterogeneous (Bernhard et al., 1987) and includes a variety of hard and soft-shelled forms representing widespread and
endemic species (Gooday et al., 1996). Subfossil foraminifera from sub-ice shelf habitats have also been studied in sediment cores, where the multi-proxy approach has allowed robust interpretation of past conditions (Kilfeather et al., 2011; Majewski et al., 2018, 2020). These micropaleontological data seem to be more complete than our metabarcoding results in the case of



Rotaliida and Textulariida (Table S3), but they do not include monothalamids, which carry important ecological information (e.g., Habura et al., 2004; Lecroq et al., 2011; Pawlowski et al., 2002a, 2002b). Our study of the small subunit of the rDNA

region now allows us to extend our knowledge of foraminifera that are not preserved in the fossil record in the poorly studied glacier-proximal habitats.

The most striking feature of the KC30 and KC49 *sed*aDNA records is the significantly lower foraminiferal diversity in the glacier-proximal sediments than in the subsurface open marine sediments represented by ST and SH (Figs. 6 and 7). Except for two monothalamid species *B. flexilis* and *C. laevis*, there are no assigned foraminiferal taxa specific to the glacier-proximal

environment. Furthermore, these two are also known from more open marine environments elsewhere (Höglund, 1947; Gooday and Pawlowski, 2004). All other taxa identified in glacier-proximal sediments also occur in open-marine settings, so the glacier-proximal assemblage appears to represent a subset of foraminifera present in open-marine settings. Nevertheless, representatives of 16 genera are present in the glacier-proximal samples from KC30 and KC49 (Table S3), including 9 monothalamid genera, together with 7 + 19 OTUs of unnamed monothalamids belonging to various clades for ST + SH and 8

OTUs representing ENFOR clades (Pawlowski et al., 2011); see Table S1. Their presence in the highly restricted environment below the ice shelf and in relative proximity to the grounding line is consistent with the findings of Habura et al. (2004), who revealed unexpectedly high foraminiferal diversity, with ca. 90% of environmental DNA reads belonging to Monothalamea in McMurdo Sound (Gooday et al., 1996).

It is also important to note that among the testate Globothalamea and Tubothalamea, only a few are genetically detected in the

glacier-proximal facies (Table S3); the *G. subglobosa*, *C. wuellerstorfi*, *N. auris* and *Cornuspiramia* sp. were present in individual samples of the ST dataset, while the SH results revealed the presence of N. auris, C. antarctica and family Trochamminidae (Table S1). Interestingly, sequences assigned to *N. pachyderma* are more abundant in the glacier-proximal facies than those belong to benthic Globothalamea, reflecting the high dispersal potential of this planktonic foraminifera. Furthermore, in the ST dataset, a single but abundant OTU of planktonic *N. pachyderma* was detected in the deepest samples

in KC30 (280 cm) and in KC49 (235 cm), dated to ca. 25 Ka (Fig. 6). This presence may indicate a rich influx of microplankton, perhaps with open ocean water close to the grounding line near the time of the LGM. In addition, the very strong presence of *N. pachyderma* in the KC30 record (Fig. 6), shown by the SH data just prior to ice shelf retreat and establishment of open-marine conditions, corresponds to the microfossil record of abundant *N. pachyderma* prior to ice shelf collapse in Pine Island Bay (Kirshner et al., 2012; Totten et al., 2017). In reality, we do not observe the same signal in KC49, nor in the ST and

micropaleontological data (Majewski et al., 2020), so this intriguing interpretation remains problematic.

In summary, there appears to be no species indicative of a glacier-proximal environment. This is due to the low diversity of the glacier-proximal assemblage, which appears to represent a subset of foraminifera found in open marine facies. However, the *sed*aDNA records of KC30 and KC49 from JOIDES Trough are very different (Fig. 4). At site KC30, the SH marker revealed a stronger presence of *N. pachyderma* and Tubothalamea (Fig. 6), which may reflect less restricted conditions. If not



due to incomplete records, this suggests considerable faunal variability in glacier-proximal benthic foraminiferal communities, which was not detected by fairly consistent micropaleontological results (Majewski et al., 2018, 2020).

**4.3 Does the length of marker matter?**

Several physicochemical factors present in natural environments, i.e., UV radiation and the hydrolysis process, as well as the biological activity of bacterial DNase, contribute to the degradation of DNA structure, causing its fragments to become shorter

and shorter with time (Blum et al., 1997; Levy-Booth et al., 2007) and core depth (Armbrecht et al., 2021). In order to enable analysing of more degraded material and to improve our reconstruction of past foraminiferal assemblages, we designed a new primer, s14F1_SH, which allows the amplification of shorter DNA fragments than the routinely used ST marker.

Our results confirm that amplicon length has a direct impact on the quantitative analysis of metabarcoding data. Overall, after filtering, we detected almost 2.5 times more OTUs with the newly designed SH than with the ST marker, i.e., 986 vs. 397 (Fig.

5), which do not always overlap (Fig. S1). This increased effectiveness of the ultra-short SH marker is manifested by significantly higher values of the alpha diversity indices (Fig. 3). Importantly, with the exception of the Simpson index, all indices show significantly higher values for the SH marker not only in subsurface, but also in surface samples, suggesting its higher effectiveness in analysing fossil, i.e., degraded DNA, but also modern DNA. The SH marker also allows a better differentiation between OTUs from different sediment types and from different cores, but mainly from subsurface samples

(Fig. 4).

The higher performance of the ultra-short SH marker could be explained by several factors. First, the newly designed forward primer can amplify a wider range of foraminiferal taxa, especially in the case of monothalamids. Indeed, only up to 15 monothalamiid OTUs are detected by the ST marker in glacier-proximal samples, whereas up to 69 OTUs are detected by the SH marker (Table S1). The SH marker also appears to perform better than ST at detecting planktonic foraminifera and

Tubothalamea (Fig. 5). Some species present in the SH dataset are absent from the ST dataset (e.g., *Astrononion echolsi*; Table S3). However, each of these species could be amplified using the ST primers, suggesting that primer specificity is not the real cause of the increased number of OTUs.

The most plausible reason for the quantitative difference between ST and SH markers is the ability of the latter to amplify very short fragments of highly degraded DNA. The usefulness of short barcodes, even <100 bp has been demonstrated on several

occasions, e.g., for plant metabarcoding using the ultra-short trnL marker (Taberlet et al., 2007; Mallott et al. 2018). It has also been shown that some foraminiferal species can only be detected in sediment samples when targeted using species-specific fragments (Lejzerowicz et al., 2013). However, our study does not clearly show that the proportion of SH and ST metabarcodes changes with sediment age.

Furthermore, the decreasing length of DNA barcodes may reduce their taxonomic resolution and lead to misidentifications, as

illustrated in our data by the assignment of some SH OTUs to tropical genera such as *Borelis* or *Planoperculina* (Table S3). The choice of marker used for metabarcoding should therefore be a compromise between the ability to amplify degraded DNA



and taxonomic resolution. In view of our results, the SH marker has the potential to become a new standard for foraminiferal paleogenomics. However, its taxonomic resolution needs to be evaluated and its performance tested in other environments.

**4.4 Complementarity of *sed*aDNA and micropaleontological records**

When comparing the metabarcoding results with the paleontological record (Table S3), the low degree of overlap is striking. The discrepancy is due to the abundance of testate forms, i.e., Textulariida, Rotaliida and Tubothalamea in the fossil data, as reported by Majewski et al. (2020), and the dominance of soft-walled Monothalamea in the *sed*aDNA data.

The paucity of fragile monothalamids in the fossil record is well known and established (Gooday et al., 1996). More intriguing is the low abundance of Textulariida, Rotaliida and Tubothalamea (Table S3) in the metabarcoding data. Excluding surface
samples, there is a limited *sed*aDNA record of Textulariida in open-marine samples, which dominate the microfossil record, and sparse *sed*aDNA record of Rotaliida in glacier-proximal samples, despite abundant tests of calcareous foraminifera present in the same samples (Majewski et al., 2020). In fact, only 1+11 (ST+SH) OTUs representing agglutinated Globothalamea (*Reophax subfusiformis* and *Arenoparrella mexicana*) were recognized in subsurface open marine samples (Table S1), and only in samples from 120 cm depth in KC30 and KC49, which were directly adjacent to the layer representing ice-proximal
sediments. The OTUs represented by the rotaliids *C. wuellerstorfi*, *Epistominella* sp., *N. auris*, *Bolivinellina pseudopunctata*, *G. subglobosa*, *Stainforthia* sp. and Tubothalamea: *Cornuspiramia* sp., *C. antarctica*, *Cyclogyra* sp. and *Spirophtalmidium* sp. were more abundant in subsurface samples, but similar to Textulariida mostly in single samples, while for many species their subfossils were present throughout large parts of cores KC30 and KC49 (Fig. 2). The Tubothalamea species detected by *sed*aDNA are actually not recognized in the fossil record (Table S3). Only the planktonic *N. pachyderma* is widespread in the
*sed*aDNA record, but a second planktonic species (*Globorotalia scitula*) is also noted in the SH results, but its presence in the Ross Sea is doubtful and it was not identified in the fossil record.

A general inconsistency between the fossil and molecular record could be due to the random nature of the PCR (Vosberg, 1989) and different amplification efficiencies due to different strengths of DNA binding depending on the lithology of the sediment. It is also possible that other natural factors, such as significant genetic polymorphism (Weber and Pawlowski, 2014)
and highly variable numbers of rDNA copies at different life stages and between different species (Weber and Pawlowski, 2013), may also bias the *sed*aDNA results. However, the striking inconsistency between fossil and molecular records found in this study (Table S3) seems more likely to be due to more specific causes. It could be due to natural sedimentary and diagenetic processes resulting in increased microfossil diversity due to reworking and selective preservation of subfossil tests. However, this possibility is unlikely as the fossil assemblages are consistent between sites and reworking of foraminiferal tests from
older strata does not appear to have occurred at sites KC30 and KC49 (Majewski et al., 2020). In addition, radiocarbon dating of foraminiferal tests confirms that the calcareous assemblage is in situ in glacier-proximal sediments (Prothro et al., 2020; Majewski et al., 2020). Selective preservation of subfossil assemblages can be also an issue, but we observe an underrepresentation rather than an overrepresentation of genetically identified Textulariida and Rotaliida.



Alternatively, the strong underrepresentation of rotaliids in the *sed*aDNA results from the glacier-proximal samples could be

explained by strong binding of DNA fragments to carbonate grains, to the point where they are difficult to extract or the DNA is highly degraded (Barton et al. 2006; Levy-Booth et al., 2007; Freeman et al., 2023). Good preservation of resilient calcareous specimens, as observed in the fossil record, may further enhance these processes. However, it is still unclear why DNA from fragile monothalamids, which dominate the *sed*aDNA results (Fig. 6), is better represented than agglutinated Textulariida in the open marine facies. It is possible that monothalamid DNA, being released quickly due to the delicate nature of their tests,

binds much more rapidly to sediment grains before bacterial DNases intensify their destructive activity (Blum et al., 1997; Levy-Booth et al., 2007) than it is in the case of robustly testate Rotaliida and Textulariida.

To sum up, although metabarcoding is increasingly used to study modern (Li et al., 2023; Nguyen et al., 2023) and past (Pawłowska et al., 2014; Nguyen et al., 2023) foraminiferal diversity, it is important to keep in mind that *sed*aDNA analysis and micropaleontological results can be highly divergent (Lejzerowicz et al., 2013; Pawłowska et al., 2014), but at the same

time highly complementary. By combining these two approaches, it is possible to reconstruct a more complete and ecologically meaningful diversity of foraminiferal assemblages. The advantage of the *sed*aDNA metabarcoding method is particularly important in marginal marine environments, where the fragile Monothalamea are dominant (Gooday et al., 1996; Nguyen et al., 2023).                                                                                                               (1)

## 5 Conclusions

In this study, high-throughput sequencing of *sed*aDNA is used to improve the understanding of foraminiferal communities inhabiting open marine and glacier-proximal environments of the western Ross Sea, mainly by adding the record of abundant and diverse monothalamids not preserved in the fossil record. By using the newly designed forward primer s14F1_SH, which allows amplification of DNA fragments that are ca. 50 bp shorter, we were able to detect higher diversity in surface and subsurface samples and discriminate between foraminiferal assemblages from different sediment types and different cores

better than with the standard approach. Thus, the newly designed ultra-short marker appears to be more accurate for paleoecological studies.

Our results, showing a consistent absence of a foraminiferal DNA signal in the tills, suggest an absence of their DNA in sediments overridden and reworked by advancing ice sheets during the last glaciation. Foraminiferal assemblages from the open marine environment show significantly greater alpha diversity than sediments deposited on the slopes of a grounding

zone wedge proximal to the grounding line. The metabarcoding method appears to be particularly useful in restricted marine environments, such as proximal to the grounding line, where fragile monothalamids predominate. Foraminifera surviving in such an environment represent a subset of the species present in open marine facies.

At the same time, the *sed*aDNA records from sites KC30 and KC49, which are located along the Last Glacial Maximum grounding zone wedge foreset in the JOIDES Trough, are significantly different. If not due to under sampling, this observation

suggests considerable variability in glacier-proximal foraminiferal communities. Interestingly, this variability is not reflected



in the micropaleontological data, which diverge strongly from the *sed*aDNA results. These two approaches are highly complementary and, when combined, provide enriched information on past biodiversity.

*Data availability.* Data used for this study will be submitted to public data repository and can be found in the Supplement.


*Author contributions.* ED and WM designed the study and participated in the fieldwork. JBA organized the field work. ED performed the laboratory work under the supervision of MB and DP and analysed the data. IBA and NLN performed bioinformatic and statistical analyses. JP helped to interpret the results. ED and WM drafted the paper, Ed prepared the figures. All authors participated in the revision of the first draft.


*Financial support.* Fieldwork for this study has been supported by National Science Foundation (grant no. ANT-1246353 to John B. Anderson), further analyses by Polish National Science Centre (grant no. NCN-2015/17/B/ST10/03346 to Wojciech Majewski).

*Competing interests.* The contact author has declared that none of the authors has any competing interests.

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

Table 1. Locations with cores and samples details. GZW - Grounding Zone Wedges, MSGL -Megascale Glacial Lineations; sedimentary facies followed by Prothro et al. (2018, 2020).





| Core number | Morphological features | Latitude (S) | Longitude (E) | Water depth (m) | Core length (cm) |
|---|---|---|---|---|---|
| KC03 | MSGL | 76°12.973' | 164°53.170' | 838 | 269 |
| KC04 | GZW topset | 76°04.688' | 170°20.057' | 597 | 200 |
| KC18 | GZW topset | 75°52.932' | 179°32.764' | 488 | 270 |
| KC30 | GZW foreset | 74°26.863' | 173°22.787' | 536 | 303 |
| KC49 | GZW foreset - toe | 74°22.382' | 173°34.827' | 541 | 258 |

Table 2. OTUs and reads ranges and average values for samples from different types of sediments. Results are shown for standard (ST) and short (SH) primer pairs, n – number of samples. The ST sample from 200 cm depth in KC49 ST was not used for calculating the averages.

| Sediment type | n | Minimal value | | Maximal value | | Average value | | Average OTUs/reads of monothalamids per sample | |
|---|---|---|---|---|---|---|---|---|---|
| | | OTUs | Reads | OTUs | Reads | OTUs | Reads | OTUs | Reads |
| **ST** | | | | | | | | | |
| All positive samples | 17 | 1 | 10 288 | 151 | 181 295 | 36.4 | 88 902 | 21.2 | 63 540 |
| Surface open-marine | 5 | 59 | 38 175 | 151 | 139 569 | 90.4 | 75 003 | 49.6 | 56 150 |
| Subsurface open-marine | 6 | 8 | 10 288 | 36 | 124 643 | 23 | 73 604 | 16.2 | 57 707 |
| Glacier proximal | 6 | 1 | 24 784 | 8 | 181 295 | 4.7 | 115 784 | 2.7 | 75 532 |
| Subglacial till | 9 | 0 | 0 | 0 | 0 | 0 | 0 | 0 | 0 |
| **SH** | | | | | | | | | |
| All positive samples | 20 | 3 | 50 682 | 116 | 295 797 | 83.8 | 135 806 | 57 | 101 429 |
| Surface open-marine | 5 | 110 | 78 741 | 382 | 187 804 | 208.6 | 109 574 | 145.4 | 79 801 |
| Subsurface open-marine | 6 | 39 | 62 142 | 116 | 295 797 | 77 | 147 353 | 53.3 | 92 241 |
| Glacier proximal | 9 | 3 | 50 682 | 43 | 316 573 | 19 | 142 680 | 9.7 | 119 570 |
| Subglacial till | 9 | 0 | 0 | 0 | 0 | 0 | 0 | 0 | 0 |



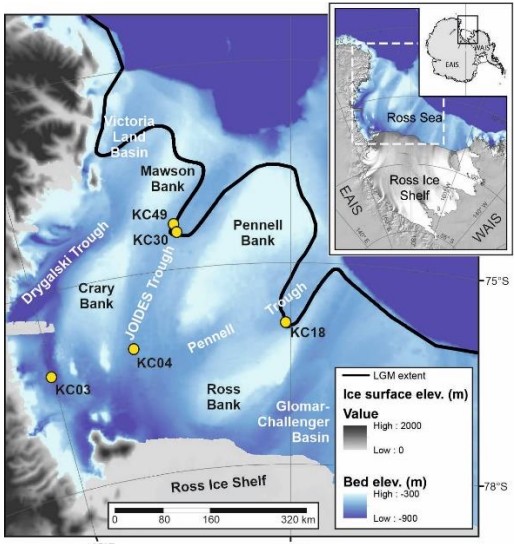

**Figure 1: Location map of the study area. Last Glacial Maximum (LGM) extent (black line) in the northern parts of JOIDES, Pennell troughs and Victoria Land Basin from Prothro et al. (2020). EAIS - East Antarctic Ice Shelf, WAIS - West Antarctic Ice Shelf.**




**Figure 2: Number of reads and OTUs for ST and SH primer pairs plotted against environmental conditions (Prothro et al., 2020)**
**and foraminiferal microfossil results (Majewski et al. 2020) in cores collected from the western Ross Sea (Fig. 1). Sample positions**
**in the cores are indicated by black arrows. Radiocarbon calibrated ages (in red) according to Prothro et al. (2020) are all measured**
**on foraminiferal tests. Note the presence of the *sed*aDNA signal only in the top samples taken from open marine sediments overlying**
**the tills and throughout the marine records in cores KC30 and KC49.**





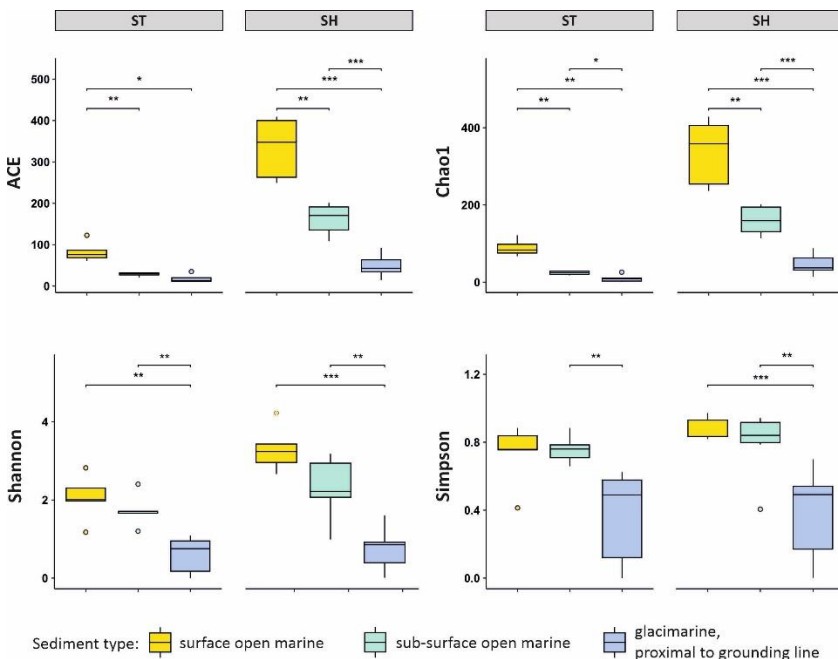


**Figure 3: Normalized alpha-diversity indices, including community richness (ACE index, Chao1) and diversity (Shannon, Simpson). Horizontal bars indicate significant differences (Wilcoxon tests, *P < 0.05. **P < 0.01 and ***P < 0.001).**

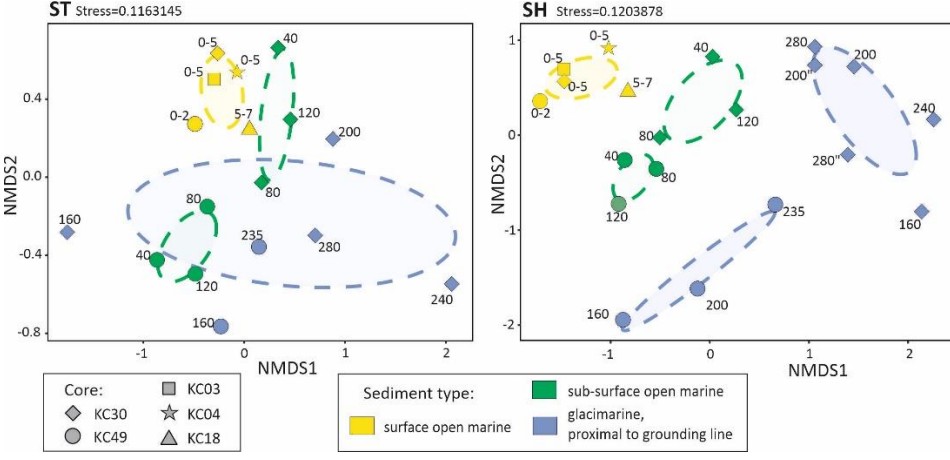


**Figure 4: Structure of foraminiferal communities derived from the *sed*aDNA approach using nonlinear multidimensional scaling based on the Bray-Curtis distance similarity coefficient for the ST and SH datasets. Stress value is displayed on the plot.**



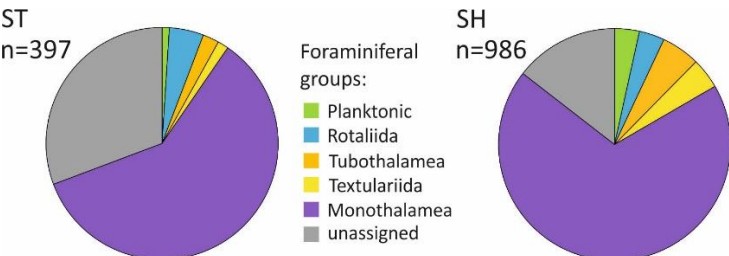

**Figure 5: Proportion of OTUs of different foraminiferal taxonomic groups detected using standard (ST) and short (SH) primer pairs, n – number of OTUs in the ST/SH dataset. OTUs with less than 10 reads in a single sample excluded.**





**Figure 6: Number of OTUs, reads (only OTUs with ≥10 reads in a single sample are shown) and percentages of OTUs of different foraminiferal groups plotted against sediment type according to Prothro et al. (2020). Samples from cores KC30 and KC49 are marked by the bars, radiocarbon ages (Prothro et al., 2020) are marked in red.**



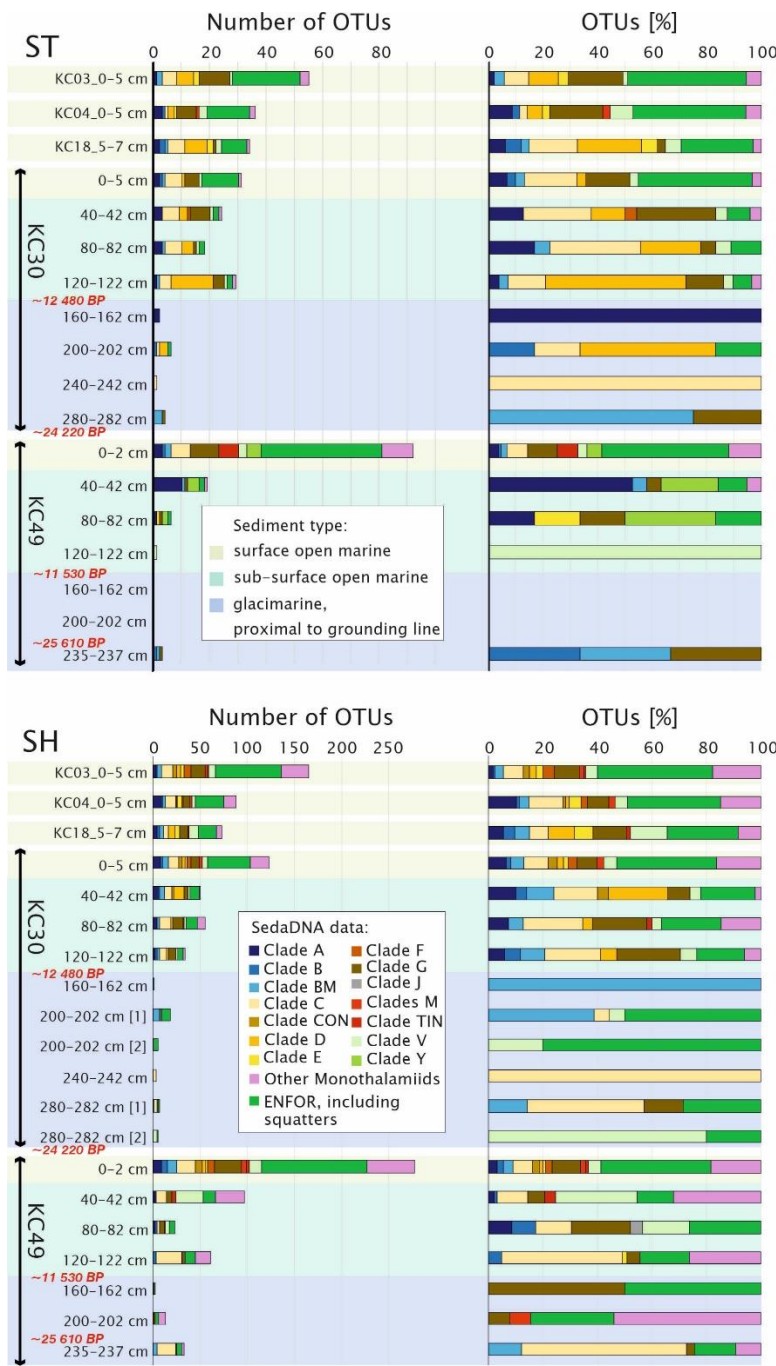

**Figure 7: Taxonomic composition of monothalamid foraminifera sequenced in samples arranged by sediment type (Prothro et al., 2020). Samples from cores KC30 and KC49 are marked by bars, radiocarbon ages (Prothro et al., 2020) are marked in red. OTU numbers and percentages (only OTUs with ≥10 reads in a single sample are shown) are grouped according to the clades identified**

**within the monothalamids (Pawlowski et al., 2011). ENFOR stands for ENvironmental FORaminifera, corresponding to clades known only from environmental sequencing (Pawlowski et al., 2014).**