# Peer review of "Sedimentary ancient DNA insights into foraminiferal diversity near the grounding line in the western Ross Sea, Antarctica"

_EGUsphere, 2024_

## Referee Comment (RC2)

[referee-annotated manuscript omitted]

---

## Author Response (AR1)

Dear Editor,

Thank you and the reviews for their comments and great feedback! Below are our answers to reviewers comment.

As for the technical issues, the supplementary Table S1, showing some source data, is a large excel file that should stay in this format. We hope it is possible, if not we will find a public repository for it.

With the kindest regards,

Ewa Demianiuk, Wojciech Majewski and co-authors

**ANSWERS RC1:**

**Firstly, the manuscript would benefit from a more refined structure, as the introduction does not fully explain the rationale for certain methodological choices—specifically, the need for a shorter barcode, which later becomes a focus of the discussion. This omission creates a somewhat fragmented flow and shifts the balance of the manuscript, as significant introductory material about sedaDNA methodologies and barcode design only appears later in the discussion section. A more focused introduction that fully describes the study's objectives would establish a clearer narrative, reinforcing the purpose of the methods and findings. In brief, it's essential to introduce the main findings that are later discussed (and even appear in the abstract).**

Response: Thank you for pointing this out. We explained the motivation for testing the shorter barcode in the introduction and added this as another aim of the study.

**While the results section is accompanied by effective figures, the text itself relies heavily on subjective descriptions and lacks specific quantitative data. Terms like "somewhere in between" (L. 207) and "very small" (L. 237) are subjective and would be improved with precise values, enabling a clearer interpretation of findings. A more quantitative approach to presenting results would better support the study's conclusions.**

Response: Thank you, we deleted the first and corrected the second.

**Furthermore, the statistical methods, while mentioned in figure captions, are not adequately explained within the main text. Including detailed explanations of each statistical test, the conditions under which they were applied, and whether they were one- or two-sided, would enhance the transparency and reproducibility of the study's analytical approach.**

**The methods section is generally well-constructed, but further elaboration on statistical analyses would strengthen the scientific rigor. Clear descriptions of each test and a justification for their selection would improve the robustness of the analysis. Expanding this section would also improve reproducibility, allowing future researchers to better follow the study's procedures.**

Response: Thank you for the feedback. We have provided detailed explanations of the statistical methods in the M&M section and we have revised the main text to include detailed descriptions of each statistical test and related package.

In the revised version we have provided more details for statistics sections.

**Lastly, the manuscript would benefit from a thorough proofreading by a native English speaker. At times, certain phrasing or wording makes it difficult to discern the authors' intended meaning. While such a review could be completed following major revisions, it would greatly improve the manuscript's readability**

Response: The final revised manuscript has been read and corrected by a native speaker.

**Technical Corrections**

**L30-31: The connection between Arctic warming and Antarctic ice sheets is not clearly articulated. It feels abrupt, with no clear link.**

Response: Thank you. For simplicity, Arctic warming has been removed.

**L37: "More advanced" is too vague—does it imply previous techniques were not advanced? Being from the sedaDNA field myself, I understand that you might mean "recently developed" or "innovative" but you need more care with your wording choice.**

Response: We changed it to "recently developed". Thank you.

**L49-51: Monothalamids are hard to track in fossil records, and sedaDNA provides an alternative means, but what role do monothalamids play in the ecosystem? Why are they of interest? A brief introduction to their ecological significance would provide context.**

Response: Sentence "Monothalamid foraminifera are particularly well represented in marine restricted environments such as fjords, including environments close to the glacier fronts (Majewski, 2010; Korsun et al., 2023), which was also confirmed by metabarcoding analyses (Nguyen et al., 2023)." from lines 51-53 has been placed earlier on.

**L61-63: There is no clear mention of the need for a new mini-barcode, which is emphasized in the abstract as a major result. Why was a shorter barcode necessary?**

Response: Thank you, we added a relevant paragraph to the Introduction.

**L65-68: I find this sentence difficult to follow. Consider rephrasing for clarity.**

Response: We changed this sentence to: "Although attempts have been made to assess diversity of Antarctic foraminiferal using metabarcoding (Habura et al., 2004; Pawlowski et al., 2011; Li et al., 2023), this is the first time when Southern Ocean subsurface sediments has been targeted and one of only a few analyses of sub-surface sediments worldwide (Lejzerowicz et al., 2013; Pawłowska et al., 2014, 2016, 2020a, 2020b; Szczuciński et al., 2016)."

**L70-73: The stated goals don't mention developing a shorter marker, yet this is presented as a significant outcome in the abstract and discussion. Clarify if this is a primary aim or a secondary result.**

Response: As stated above, this has been added as one of the aims in the introduction.

**L151: Unusual spacing between "and" and "3uL."**

Response: This has been corrected.

**L169: A fixed 90% coverage threshold affects the markers differently, as the shorter marker would allow more mismatches. Please clarify this point.**

Response: A fixed 90% coverage threshold impacts markers differently because shorter markers inherently tolerate a greater proportion of mismatches or gaps while still meeting the 90% similarity criterion. The differential effect can influence the taxonomic resolution and accuracy of sequence assignments across markers of varying lengths. Based on the length of 2 markers, the 90% identity with allowing up to 7 mismatches or gaps and 90% coverage was fixed for BLAST to reduce the differential effect.

**L195: The notation "135 80" is unclear—is this 13,580, 13 580, or two separate values?**

Response: Thank you for pointing out that. It is corrected now.

**L195-196: The sentence is unclear regarding replicates for all samples except those at 200 and 280 cm. Please clarify.**

Response: The sentence "SH samples in KC30 from 200 and 280 cm were analysed in two replicates." has been moved to the method section where it fits better.

**L207: "As significantly as" may suggest a statistical difference. Please ensure accurate wording to avoid misinterpretation.**

Response: Thank you. We modified the sentence and avoided using "significantly" without a statistical test.

**L216: Does "they" refer to datasets or patterns? Clarify what is being described as "more scattered."**

Response: "They" have been changed to "community composition".

**L218: "Slightly different" is subjective. Aim for a more precise description (other instances of subjective terms were noted throughout the results but not all listed here—please review for accuracy).**

Response: We changed it to "clusters for surface and subsurface samples, that do not overlap"

**L222: "Tend to form"—do all samples follow this pattern, or only some? Provide specific values.**

Response: We changed the sentence as follow: "The surface open-marine samples from different cores form a single tight cluster." Thank you it was a vague statement, indeed.

**L237: "Very small numbers" is vague—specify.**

Response: We changed it to "small numbers, i.e., < 3%"

**L260: "Highly irregular"—what is considered regular in this context? Constant over time or without variation?**

Response: We added "with depth".

**L260-275: The description of SH and ST detection of OTUs is difficult to follow within this paragraph. Consider restructuring for clarity.**

Response: Thank you, we introduced some changes in this paragraph to clarify.

**L280-304: This section lacks discussion or interpretation of results and instead offers a lengthy introduction to sedaDNA preservation.**

Response: This part is shorter now and moved to the Introduction as the motivation for aiming to develop the shorter barcode.

**L286: Unusual spacing between "sediments." and "To determine."**

Response: It has been corrected.

**L310: When discussing sediment mixing and dilution, does this imply mechanical damage to DNA fragments or dilution due to sedimentation rate? Clarify.**

Response: Thank you, more of the first, i.e., mixing/reworking. We slightly changed this sentence.

**L321: Could you report DNA concentrations post-extraction or gel bands after PCR amplification?**

Response: We checked DNA concentration after extraction for each sample only to make sure that there was genetic material for further analysis. Measurements were conducted with two different machines (one part of this study was performed after a lengthy break), which is the reason we decided to not include the numbers.

**L323-326: Shotgun sequencing could confirm DNA preservation. Even without sequencing, a Bioanalyzer or TapeStation could verify fragment distribution, which may confirm the need for a shorter marker.**

Response: Thank you, we added this suggestion as one of the recommendations.

**L377-409: This key discussion point about shorter markers was not introduced earlier—consider integrating into the introduction.**

Response: Yes, it has been done.

**L378-382: The rationale for a shorter marker should appear in the introduction.**

Response: As above.

**L404-405: Have you checked databases for potential sequence overlaps? Your taxonomic assignments should follow the Last Common Ancestor (LCA) principle. Therefore, the use of a shorter fragment would lead to potential lower taxonomic resolution (i.e., the inability to differentiate between two species) rather than misassignment. If all species within Borelis and Planoperculina are exclusively tropical and show no sequence overlap with the ST marker what are the closest relatives present in the databases?**

Response: Thank you for pointing this out. We agree with the reviewer's comment that SH provides a lower taxonomic resolution since no OTUs were assigned to tropical species in the ST. The assignment for both ST and SH markers was done using BLAST best hit as in M&M (90% similarity and 90% coverage and 7 gaps or mismatches). As a result, some SH sequences were assigned to tropical species. In the main text, we have provided and discussed criteria for avoiding misidentification.

**L453: Unusual spacing between "(2023)." and "(1)."**

Response: Corrected.

**L454-473: Conclusions should reflect the study's aims, with better alignment between introduction and final conclusions.**

Response: Yes, Introduction has been expanded.

**Figures:**

**The figures are generally clear. You may wish to consider scaling the y-axis with depth, allowing for easier identification of core coverage. Also, note that only two dating points are provided for the entire record, which may limit chronological confidence.**

Response: Thank you. In Fig. 2, core-depth axes are now labeled. Figure captions for Figs, 6 and 7 are updated.

**ANSWERS RC2:**

**L44: Seidenstein et al. 2024; Marine Micropalaeontology**

Response: Thank you, we added this reference.

**L330: Dameron et al. 2024; J Micropalaeontol.**

Response: Thank you, we added this reference.

**L342: Bombard et al. 2024 for the Miocene of the Ross Sea, and Seidenstein et al 2024 for the Plio-Pleistocene; both in J. Micropaleontol.**

Response: Thank you, we added this references.

**L363: This is a really important observation: That planktic forams can be drawn deep under ice shelves; not likely living, but their tests can be transported great distances under the ice shelves. Dameron et al. 2024 report very rare N. pachyderma in both the upper unit (Holocene) and lower unit (late Miocene in age) of RISP; the latter helping to constrain the likely late Miocene age of the till. Likewise, Leckie and Webb 1986 and Bombard et al. 2024 report multiple occurrences of very rare planktic forams in Miocene age core samples at DSDP Site 270 and IODP Site U1521.**

Response: Thank you, we highlighted this observation and added references to modern observations from Amery Ice Shelf (Hemer et al. 2007) and fossil records cited in your comment.

**L371: I'm not so sure. Bart et al., Prothero et al. and Majewski et al. have all clearly documented the presence of Globocassidulina subglobosa (or descendent G. biora) as the dominant mineralized benthic foram at and near GZWs in both the western and eastern Ross Sea. Bombard et al. and Seidenstein et al. have hypothesized that G. subglobosa is a proxy for warm CDW (or mCDW), which is why we think it's associated with the retreating GZWs. Both these studies have demonstrated the association of G. subglobosa-dominated benthic foram assemblages with incursions of warmer water planktic forams into the Ross Sea during the Miocene and Plio-Pleistocene, respectively. It would be interesting to see this type of DNA analysis conducted on the RISP cores, both the upper and lower unit (the latter is a till and likely barren of DNA but worth a try).**

Response: In that statement, we referred to our sedaDNA results. This is made clearer now.

**L414: I agree; why aren't these other taxa represented unless monothalamids so greatly dominate?**

Response: We discuss the reasons down from line 431, for Rotaliida and Textulariida specifically in lines 443-450.

**L425: This G. scitula result does not surprise me at all based on the numerous species of subpolar, temperate, and even subtropical (G.. ruber!) we have found in our Ross Sea core samples. I reported G. trilobus in the basal Miocene at DSDP Site 270, as well as other warmer water species (Leckie and Webb 1986), many warmer water species documented in the Miocene of U1521 (Bombard et al 2024), and G. ruber and G. inflata in the upper Pliocene of U1523, among other subpolar species until 1.82 Ma (Seidenstein et al. 2024). Although we have documented many different planktic species, I have not yet seen scitula., but your results do not surprise me.**

Response: You are right. Scitula's DNA was noted in open-marine facies, when only a few calcareous tests survive. The sentence has been corrected to be more neutral: "but a second planktonic species (*Globorotalia scitula*) is also noted in the SH results, which presence was not identified in the fossil record."

**L461: Please briefly explain why ultra-short markers are more accurate for paleoecological studies.**

Response: Inaccurate wording; We checked it as follow: "Thus, the newly designed ultra-short marker appears to be potentially more useful for paleoecological studies."

**Fig 2: Cores KC03, KC04, and KC18 bear striking resemblance to the upper unit of the cores recovered at RISP (Ross Ice Shelf Project of the late 1970s); specifically, agglutinated benthics dominate the top of the cores (green unit here) with very rare calcareous benthic specimens and very few planktic Neogloboquadrina pachyderma, overlying an unconformity with sparse calcareous benthic foram and diatom assemblages of mixed Miocene ages in the lower unit at RISP (brown unit here) (please see Harwood et al. 1989, Mar. Micropaleontology; Dameron et al. 2024, J. Micropalaeontology). However, the ages of the till here (brown unit) are clearly not Miocene. Have DNA analyses of the RISP cores been investigated?**

Response: The dominance of agglutinated foraminifera in upper open-marine sections of cores from deeper (> ca. 500 m) parts of the Ross Sea is a common feature (Majewski et al. 2018, 2020), regardless of the type of underlying sediments. It is simply a consequence of elevated CCD. DNA studies should be conducted on freshly collected material. Samples are preferred to be collected directly after coring or from frozen cores. For this reason, RISP cores are rather not suited for such analyses, if we understood the last sentence of this comment correctly.

**Any diatom biostratigraphy or diatom assemblages from the lower (brown and blue) units of these cores?**

Response: No, no diatom data.

**Fig. 3: Please define what the ST and SH datasets are in this figure caption.**

Response: Defined now in the figure caption.

**Fig. 4: Please define what the ST and SH datasets are in this figure caption.**

Response: As above.

---

## Author Response (AR2)

Dear Editor,

Thank you for the final comments. We corrected the manuscript according to the comments.

**- Sections 1.1 and 1.2 should be moved to 'Materials and Methods'**

Done
**- Cruise NBP1502A – please specify timing of the cruise and research vessel used**

The details are dded at the beginning of section 2.2.
**- The sentence regarding replicates remains unclear (see lines 161 and 184). Please, rephrase.**

It is now changed into "Two replicates were analysed for SH in samples 200 and 280 cm from KC30." At the end of section 2.4.
**- Fig. 5 is cited before Figs. 3 and 4; Fig. 7 is cited before Fig. 6. Please, revise the figure numbers according to the order in which they are cited in the text**

Early citation of Fig. 5 is removed. Figs 6 and 7 renumbered.

Moreover, the *Data availability* statement has been updated. And the doi location of Table S1 indicated in the Supplement.

With the kindest regards,

Wojtek Majewski and co-authors